# Potential Factors Influencing Complete Functional Recovery in Traumatized Unowned Cats with Orthopedic Lesions—A Cohort Study

**DOI:** 10.3390/vetsci11020059

**Published:** 2024-02-01

**Authors:** Francesco Ferrari, Liliana Carnevale, Federica Alessandra Brioschi, Jessica Bassi, Davide Danilo Zani, Stefano Romussi, Luigi Galimberti, Damiano Stefanello, Sara Rioldi, Luigi Auletta

**Affiliations:** 1Department of Veterinary Medicine and Animal Sciences (DIVAS), University of Milano, Via dell’Università 6, 26900 Lodi, Italy; francesco.ferrari@unimi.it (F.F.); lilianacarnevale@tiscali.it (L.C.); federica.brioschi@unimi.it (F.A.B.); jessica.bassi@unimi.it (J.B.); davide.zani@unimi.it (D.D.Z.); stefano.romussi@unimi.it (S.R.); sara.rioldi@guest.unimi.it (S.R.); luigi.auletta@unimi.it (L.A.); 2Agenzia di Tutela della Salute—ATS Città Metropolitana di Milano Distretto Veterinario Alto Lodigiano, Sant’Angelo Lodigiano, 26900 Lodi, Italy; lugalimberti@ats-milano.it

**Keywords:** unowned cats, high energy trauma, traumatic orthopedic injuries, panleukopenia, functional recovery

## Abstract

**Simple Summary:**

Unowned free-roaming cats are companion cats who are lost or abandoned and who are living individually or in a feline group. The status of unowned cats influences the incidence of pathology compared to owned cats. Trauma is one of the most frequent causes of hospitalization in this population. Little is known regarding the outcome of traumatic orthopedic injuries in these patients. The present study aimed to identify the factors that could influence the ability to move well and maintain normal behaviors (complete functional recovery, CFR) in traumatized unowned cats with orthopedic lesions. All cats referred by the veterinary public service to a veterinary teaching hospital, over a three-year period, were enrolled. Various clinical variables were retrospectively evaluated. Forty-eight unowned cats were included in the study, of which thirty-four had CFR. Higher body weight and a longer time from trauma to therapeutic intervention were associated with CFR. These results showed that lighter cats who survived to presentation experienced more severe consequences following blunt trauma and emergency procedures were associated with a poorer prognosis. Five of the fourteen patients who did not reach CFR died due to feline panleukopenia. Overall, unowned cats with traumatic orthopedic injuries showed a good prognosis.

**Abstract:**

The management of unowned cats is an emerging problem, with public institutions and citizens’ concerns regarding their care and arrangement. Little is known regarding the outcome of traumatic orthopedic injuries in these patients. Indeed, complete functional recovery (CFR) should be the goal of treatment for return to their original location or adoption. The aim was to identify clinical factors influencing CFR in traumatized unowned cats with orthopedic lesions. This category of cats referred by the veterinary public service over three years was enrolled. Various clinical variables were retrospectively collected from the medical records and evaluated by nominal logistic analysis. Forty-eight unowned cats were enrolled, with a median estimated age of 24 (1–180) months and a body weight of 3 (0.7–5) kg. Thirty-four (71%) patients reached CFR. Estimated age, body weight, time from trauma to therapeutic intervention, spine involvement, presence of comorbidities, hospitalization time, and the radiographic score results were significantly associated with CFR. A longer time to therapeutic intervention seemed to be associated with a better outcome. Probably, cats severely traumatized did not live long enough to be evaluated and treated. Lighter cats experienced more severe consequences following blunt trauma. Younger and lighter cats bore a higher risk of panleukopenia-related death.

## 1. Introduction

Unowned free-roaming cats are synanthropic animals, whose needs are directly or indirectly supplied by humans, living near human-populated areas, individually or in groups [1]. The Italian law regulates the management of unowned cats, recommending that local public health veterinary services take care of unowned free-roaming cats, ensuring their health status and survival conditions, and mandating euthanasia only in case of untreatable pathology [2,3]. The Italian population intensely perceives the role of public health service, feeling major concerns about unowned cats’ welfare and public health issues [4]. In this context, the Veterinary Teaching Hospital of the University of Milan (VTHUM) works as second opinion veterinary practice providing care and hospitalization for traumatized unowned free-roaming cats, cooperating with the veterinary public service.

Trauma has been reported as a frequent cause of hospitalization and death in unowned free-roaming dogs and cats [5,6]. Indeed, it has been shown that acute dyspnea of traumatic origin is significantly more frequent in unowned free-roaming than owned cats, highlighting the marked differences in lifestyle and incidence of traumatic pathologies in these two feline populations [7]. Nonetheless, owned cats were more frequently presented for dyspnea due to different causes, e.g., cardiogenic and neoplastic, probably because of the owners’ ability to identify early signs of illness [7]. 

Nevertheless, little is known about the clinical characteristics and outcome of traumatic orthopedic injuries in the unowned feline population, in contrast to owned cats [8,9,10]. The presence of pre-existing pathologies or comorbidities, generally undiagnosed and hence untreated, in addition to concomitant traumatic soft tissue injuries, might influence the response to treatment and the outcome of orthopedic lesions in traumatized unowned cats. 

The destiny of unowned cats might depend on the final outcome after treating traumatic orthopedic injuries. A complete functional recovery (CFR) with only subtle disabilities, but with the patients able to freely move and jump, without any difficulty in feeding, urinating, and defecating, should be the desired outcome. After recovery, the possible arrangement of unowned cats includes the return to their original location, introduction into a shelter, or adoption. Return to the original location should be considered only for those patients that experience CFR. Introduction in a shelter seems to be perceived as a more ethical solution by people in comparison to returning to their original location. However, in felines, immunosuppression has been associated with chronic stress from living in shelters, and it is considered a significant factor leading to the development of upper respiratory infections or reactivation of previous infectious agents [11]. Moreover, the economic cost of their management constitutes a burden on the public administrations. Anyhow, in the absence of CFR, introduction in a shelter or adoption seems to be the only option. Adoption might be influenced by several factors, such as location, characteristics of the cat, including behavioral issues, and previous interactions with humans [12,13], and it is felt more unlikely, at least in Italy, for cats needing continuous management or medical care [13,14].

Hence, improving the knowledge about the management of traumatized unowned cats is paramount to minimize hospitalization costs, maximize the probability of CFR and then guarantee their return to their original location or adoption. The present study aimed to describe the clinical features of orthopedic lesions in traumatized unowned cats and identify possible clinical prognostic factors leading to CFR.

## 2. Materials and Methods

The electronic medical records of all unowned cats referred by the veterinary public service for trauma to the VTHUM between November 2020 and May 2023 were evaluated for enrolment in this cohort study by one author (F.F.). Inclusion criteria were the presence of at least one traumatic orthopedic lesion, medical records including signalment data, trauma type, estimated time from trauma, type of traumatic orthopedic lesions, concomitant traumatic non-orthopedic lesions, therapeutical approach, and stabilization procedures if performed, date of discharge, and clinical report of the 30-, 60- and 90-day re-evaluation from the definitive treatment. Clinical reports of the 30-, 60- and 90-day re-evaluation should have included at least the state of consciousness; data about usage and healing of the anatomical region affected by the traumatic lesions; respiratory pattern; and data about the ability of feeding, urinating and defecating. Exclusion criteria were cats referred by the public health veterinary service as unowned, but subsequently identified as owned cats via the presence of a microchip or owner claim; cats positive to antigenic tests for feline immunodeficiency virus or feline leukemia virus; cats with medical records lacking the 30-, 60- and 90-day re-evaluations. All unowned cats referred at the VTHUM are routinely tested for feline immunodeficiency virus, feline leukemia virus, and feline parvovirus.

The CFR was defined by the authors modifying a previous orthopedic subjective clinical outcome evaluation [15] as the absence of any disability that would prevent the patients from freely moving and jumping. As routinely performed for evaluating unowned cats, whenever needed, patients were freed in a dedicated consult room, and they were observed by one of the authors behind a one-way mirror. Moreover, CFR was considered not reached if mental status alterations, sight deficits, clinical or reported straining to urinate or defecate or inability to properly posture for these functions, inability to properly feed due to malocclusion, presence of any alteration in the respiratory pattern, or presence of cutaneous sores were detected at the clinical evaluation.

Variables collected for each patient were breed; estimated age—on the basis of the anamnestic and clinical information from the caregivers/public veterinary service, and on the evaluation of dentition; body weight; sex and neutering status; body condition score—BCS—on a nine-points scale [16]; trauma type, penetrating or blunt; estimated time from trauma to case referral, defined as the time elapsed from the first identification by the public veterinary service of an unowned cat symptomatic for trauma to its referral at the VTHUM; acute orthopedic lesions, categorized as fracture, luxation or fracture and luxation; if present, chronic orthopedic lesions were recorded, as well; orthopedic lesions were assigned to the acute or chronic category according to the clinical and radiographical evaluation; concomitant soft tissues and neurological traumatic lesions; anatomical region at which trauma was located, divided into head, spine, forelimb, hindlimb, pelvis, thorax and abdomen—sacro-iliac fracture and luxation were included in the spine region; whether traumatic lesions were localized to a single or multiple regions; presence of non-traumatic comorbidities; estimated time from trauma to therapy; type of therapy, classified as surgical or conservative; when surgical, whether internal of external fixation was used; need for blood transfusion pre- and post-surgery; application of esophagostomy feeding tube; hospitalization length; clinical re-evaluation performed 30, 60 and 90 days post-treatment; and whether the patient was adopted, returned to their original location or sheltered. If not already neutered, all patients were sterilized after complete recovery from the traumatic events, prior to adoption or returning. According to the Italian regulation, approximatively 5 mm of the ear tip was removed in all unowned cats undergoing neutering. 

The clinical re-evaluations at 30 and 60 days post-treatment were considered positive when there was an improvement of clinical signs associated with the traumatic lesions identified at the presentation. A negative clinical evaluation was assigned in all other cases, and in particular when there was no improvement, e.g., usage of the affected limb; moreover, any possible post-surgical complication, such as surgical site infections, was also scored as negative. Results of ancillary tests performed, e.g., radiographic evaluation, were included in the definition of positive or negative 30- and 60-day re-evaluations. The CFR was evaluated at 90 days post-treatment, or previously whenever a clear negative outcome was recorded, i.e., death of the patient, as well as an obvious positive CFR. 

Cause of death was assessed by means of necroscopy if available; otherwise, the possible cause of death was estimated based on the medical report. A presumptive diagnosis of panleukopenia as the cause of death was emitted based, at least, on appropriate clinical signs and a fecal parvovirus antigenic ELISA test, performed either at the investigating institution or by the shelter veterinarian [17], but in the absence of a possible clinical correlation to the traumatic event as well as the therapeutic intervention.

### 2.1. Radiographic Scores

Plain radiographies in two orthogonal projections of any anatomical region clinically involved or the total body were evaluated. To summarize radiographic lesions and to test them as possible predictors of the CFR, 3 different scores were developed or adapted from previous scales. A skeletal score, ranging from 0 to 3 points, was generated by the authors (two experienced radiologists—J.B., D.D.Z—and one experienced clinician—F.F.) by evaluating the radiographic study. In particular, one point was assigned if at least a fracture or luxation affected the axial skeleton, one point was assigned if at least a fracture or luxation affected the appendicular skeleton, and one point was assigned if there was involvement of any joint. A thoracic score, spanning from 0 to 4 points, was adapted from Sigrist and *coll.* [18]. One point was assigned for the presence of each of the following abnormalities: pneumothorax or pneumomediastinum, pulmonary contusion, pleural effusion and chest wall or diaphragmatic lesion. Likewise, an abdominal score, spanning from 0 to 3 points, was developed by the authors (two experienced radiologists—J.B., D.D.Z—and one experienced clinician—F.F.) by assigning one point to each of the following abnormalities: the presence of peritoneal effusion, peritoneal gas and herniations with the exclusion of diaphragmatic hernia. The sum of the aforementioned radiographic scores created a ten-point scale, named total trauma radiographic score (TTRS). All scores were assigned by the same operator, an experienced radiologist (J.B.).

### 2.2. Statistical Analysis

Data were recorded on an electronic spreadsheet before being imported in a dedicated commercial software (JMP Pro, v. 16.0, SAS Institute, Cary, NC, USA). For descriptive statistics, continuous data are reported as mean ± SD when normally distributed and median (range) when non-normally distributed; categorical data are reported as numbers (% of the total). A probability test about proportion according to sex and neutering status was performed, hypothesizing that each of the four categories (i.e., intact male, neutered male, intact female, spayed female) were evenly represented (probability of 25%).

Cats were classified first in two outcomes (CFR vs. no-CFR) labelled as 2-CFR throughout the manuscript, then in four outcome groups (CFR, no-CFR, death from panleukemia, death from comorbidities) labelled as 4-CFR throughout the manuscript. To evaluate the effect of clinical variables and radiographic scores on the CFR, a univariate nominal logistic regression was applied. In particular, the dependent variable was the outcome (i.e., CFR and no-CFR; then CFR, no-CFR, death from panleukemia, death from comorbidities) whereas the independent variables tested were estimated age; body weight; sex; BCS; type of acute orthopedic lesions; the presence of chronic orthopedic lesions; concomitant soft tissues and neurological traumatic lesions; anatomical region at which trauma was located—for data analysis each body region was singularly labeled as yes or no according to their involvement; whether traumatic lesions were localized to a single or multiple regions; presence of non-traumatic comorbidities; estimated time from trauma to therapy; type of therapy; internal or external fixation; need for blood transfusion; application of esophagostomy feeding tube; hospitalization length; clinical re-evaluation results at 30 and 60 days post-treatment. At the univariate nominal logistic, each independent variable was tested singularly.

All variables with *p* < 0.10 at the univariate nominal logistic regression were further tested with the following post hoc analysis considering the 2-CFR. Receiver operator characteristic (ROC) tables were analyzed to obtain a cut-off value, according to the Youden index *J*; such variables were then categorized in two groups, accordingly (i.e., for body weight < 3.3 kg and > 3.3 kg). The following tests are referred to post hoc in the Results section. Categorical variables were tested with contingency tables and *chi-square* or Fisher’s *F* test, and the odds ratio (O.R.) and the corresponding 95% confidence intervals (95%CI) were calculated, as well. Continuous variables were compared between the 2-CFR categories with Student’s *t* test or Mann–Whitney’s *U* test, according to data distribution.

When considering the 4-CFR, the independent variables were compared between the four outcomes (CFR, no-CFR, death from panleukemia, death from comorbidities) with an ANOVA or a Kruskal–Wallis’s test; post hoc, a Tukey’s HSD test was applied.

Furthermore, the variables with *p* < 0.10, excluding the 30- and 60-day re-evaluations, were introduced in a multivariable logistic regression to evaluate their independent ability to predict the CFR [19](considering both the 2-CFR and 4-CFR). In the multivariable model, all significant variables were tested together. The R^2^ and the *p*-value of the whole model, and the lack-of-fitness (LoF) *p*-value were reported as an estimate of the model fit [20].

The association between the final destiny of each cat, i.e., adoption, releasing or sheltering, and the 2-CFR was explored using contingency tables and a chi-square test. The patients who died before reaching the CFR were excluded from this case scenario. Overall, the significance was set at *p* < 0.05.

## 3. Results

In total, 60 unowned cats were referred for trauma during the 3-year study period, and a total of 346 unowned cats were referred for various medical conditions; 8 patients were excluded because only soft tissue traumatic injuries were observed and no traumatic acute nor chronic orthopedic lesions were detectable, 3 patients were subsequently identified as owned cats, and 1 cat was positive for feline immunodeficiency virus. The final sample included 48 cats, all European shorthairs; the median estimated age was 24 (1–180) months and the median body weight was 3 (0.7–5) kg. Included in the sample were 22 (46%) intact males, 10 (21%) neutered males, 12 (25%) intact females, and 4 (8%) spayed females; intact males were overrepresented (*p* = 0.003). Cats had a BCS of 3/9 in 6 cases (13%), 4/9 in 28 cases (58%), 5/9 in 8 cases (17%), 6/9 in 5 cases (10%) and 7/9 in 1 case (2%). Signalment data for each patient are summarized in Appendix A. All traumas were considered blunt, since in no cases penetrating wounds were detected clinically and after surgical wound exploration. Similarly, one (2%) complete transection of the distal hindlimb from the tarsus was considered the effect of a blunt trauma according to the clinical evaluation. 

The median estimated time from trauma to referral was 3 (0–60) days. Among the acute traumatic orthopedic lesions, fractures alone were detected in 25 (52%) cats, fractures associated with luxation in 11 (23%) cats, luxations alone were detected in 7 (14%) cats, whereas no acute orthopedic lesions were detectable in 5 (10%) cats. Chronic orthopedic lesions were diagnosed in 6 (12%) cats: 5 (10%) cats displayed signs of previous fracture and 1 (2%) had fractures associated with luxation; 3 (6%) had concomitant chronic and acute orthopedic lesions. Concomitant soft tissue and neurological lesions were detected in 22 (46%) cats: in particular, 8 (17%) patients displayed neurological signs, either central or peripheral; 6 (12%) had cutaneous bruises, excoriations and superficial lacerations; 5 (10%) had diaphragmatic hernia; and 1 (2%) each with urethral avulsion, abdominal hernia, and evisceration. In total, 25 (52%) patients were traumatized at a single anatomical region, whereas 23 (48%) showed multiple regions involved. Comorbidities were diagnosed in 15 cats (31%). Clinical data for each patient are summarized in Appendix A.

The median time from trauma to therapeutical intervention was 7 (0–60) days. Surgical therapy was applied in 38 (79%) cats, conservative management was applied in 9 (19%), whereas 1 (2%) cat underwent euthanasia for renal failure unresponsive to medical treatment. Among the surgical-treated orthopedic lesions, 24/38 (63%) were treated by internal fixation, 5/38 (13) were treated by external fixation, whereas other surgical treatments (e.g., limb amputation, open reduction luxation and stabilization, hemipelvectomy) were applied in 9/38 (24%). None of the enrolled patients needed blood transfusion pre-surgery, and only 1 (2%) needed it post-operatively. In total, 8 (17%) cats required the application of an esophagostomy feeding tube. Post-therapy complications were detected in 7 (15%) cats, both during hospitalization or at the clinical re-evaluation; in particular, 4 (8%) cats had failure of orthopedic implants, and 3 (6%) cats had surgical site infections [21]. Median hospitalization time was 14 (1–90) days. Clinical data for each patient are summarized in Appendix A. 

In total, 46 (96%) patients underwent radiographic examination; the 2 (4%) patients who did not undergo diagnostic imaging both displayed partial or total transection of a distal hindlimb. Of these two cats without imaging, one cat underwent immediate surgical treatment with amputation, whereas the severity of the clinical presentation of the second patient prevented performing diagnostic imaging. In total, 34 (71%) cats had thoracic radiographs performed and 36 (75%) had abdominal radiographs. Radiographic scores for each patient are summarized in Appendix A.

A total of 8 (17%) patients died before the first check at 30 days; 5 (10%) died from panleukopenia, and the remaining 3 (6%) due to comorbidities unrelated to the trauma [death from severe renal failure (1, 2%); post-mortem signs compatible with feline infectious peritonitis (1, 2%); and post-mortem signs compatible with cardiac failure for hypertrophic cardiomyopathy (1, 2%)]. At the 30-day clinical re-evaluation, 34 (71%) patients were considered positive and 6 (8.4%) were considered negative, whereas at the 60-day clinical re-evaluation, 35 (73%) patients were considered positive and 5 (10%) were considered negative, according to the grading system described. All the patients classified as positive at the 30-day re-evaluation had the same judgment at the 60-day reevaluation. The CFR was considered to be reached for 34 (71%) patients, and not reached for 14 (29%) patients. One patient classified as “positive” at the 60 days reevaluation, due to progressive improvement, did not meet the criteria to be classified as having reached the CFR. Among the patients experiencing the CFR, 3/34 (9%) were judged to have reached the CFR at a median of 58 (57–62) days. The remaining 31/34 (91%) were finally evaluated and classified as having reached CFR at a median of 93 (84–105) days. Concerning patients not reaching CFR, 8/14 (57%) died before the 30-day re-evaluation (median 19; 12–29 days), and the remaining 6/14 (43%) were finally evaluated at 92 (median 86–102) days. Clinical and radiographic variables significantly associated with the 2-CFR, and the relative statistical results, are summarized in Table 1. Non significantly associated variables and corresponding *p* values are summarized in Table 2.

The body weight cut-off resulted in 3.3 kg at ROC analysis with 2-CFR. The median body weight of cats experiencing CFR (3.15, 0.7–5) was significantly higher (*p* = 0.036; Appendix A) than cats with a negative outcome (1.6, 0.9–4.1). 

The time from trauma to therapeutic intervention cut-off resulted in 7 days at ROC analysis with 2-CFR. The median days elapsed for cats experiencing CFR (8, 0–60) were significantly longer (*p* = 0.039; Appendix A) than for cats with no CFR (4, 0–31). Moreover, median days elapsed from trauma to therapeutic intervention for cats with (5, 0–60) and without concomitant lesions (9, 0–60) were compared, and no significant difference could be detected (*p* = 0.10). 

The skeletal score cut-off resulted in a score of 2 at ROC analysis with 2-CFR. Nonetheless, no significant differences in the median skeletal score could be detected between cats experiencing and not experiencing CFR (*p* = 0.13).

Due to the absence of either thoracic or abdominal radiographs, the TTRS was obtained for 30 patients. The TTRS cut-off resulted in a score of 1 at ROC analysis with 2-CFR; the association between the TTRS categories (≤1 vs. ≥2) and CFR did not result significant (*p* = 0.11). 

Due to the lower number of patients with a TTRS, the multivariable analysis with 2-CFR was repeated with and without this effect. When the TTRS was included, the goodness of fit of the multivariable model was highly significant (R^2^ = 0.84, *p* = 0.0004; LoF *p* = 0.99), and the variables significantly associated with the CFR were BCS (*p* = 0.004), the skeletal score (*p* < 0.0001) and the TTRS itself (*p* = 0.008). When the TTRS was excluded, the goodness of fit of the multivariable model was non-significant (R^2^ = 0.34, *p* = 0.052; LoF *p* = 0.53), and no independent predictors of the CFR could be detected.

Due to a relatively high percentage of patients not reaching the 90-day evaluation due to death from panleukopenia (5, 10%) or other causes (3, 6%), previously included in the not-reached CFR category, a further statistical evaluation was performed with the 4-CFR. Clinical and radiographic variables significantly associated with the 4-CFR categories, and the relative statistical results, are summarized in Table 3. Non-significantly associated variables and corresponding *p*-values are summarized in Table 2.

The Kruskal–Wallis analysis on the 4-CFR identified that cats dying from panleukopenia were younger compared to all other outcome classes (*p* = 0.04; Figure 1a), and lighter than cats experiencing CFR (*p* = 0.01; Figure 1b); death not related to panleukopenia resulted in being significantly associated with the presence of comorbidities (*p* = 0.032)—indeed, from the 3 cats not deceased from panleukopenia, 1 was diagnosed with renal failure, 1 with feline infectious peritonitis, and 1 with cardiac failure for hypertrophic cardiomyopathy. The TTI did not differ between the 4-CFR (*p* = 0.16). The involvement of the spine did not result in being associated with the 4-CFR (*p* = 0.06). The TTRS did not differ between the 4-CFR (*p* = 0.06).

In the multivariable model on the 4-CFR, the goodness of fit was highly significant (R^2^ = 1.0, *p* < 0.0001; LoF *p* = 1.00), and body weight, time to therapeutic intervention, spine involvement, hospitalization time and TTRS retained their significance as independent predictors for the association with the specified four outcomes (*p* < 0.0001), whereas estimated age and presence of comorbidities were no longer significant.

Concerning the arrangement considering the 2-CFR, 26 patients (22 CFR and 4 no-CFR) were adopted after the discharge. In total, 5 patients were received in a shelter (3 CFR and 2 no-CFR). The remaining 9 patients, all with CFR, were reintroduced in a feline group. No association could be detected between the CFR and the final destiny of the patients enrolled in this study (*p* = 0.13).

## 4. Discussion

The aim of our study was to describe the clinical features in traumatized unowned cats, and their influence on the CFR. Despite the several variables recorded and examined in this peculiar sample population, those resulting independent predictors of the CFR at the multivariable analysis when considering the 2-CFR were BCS, the skeletal score and the TTRS, whereas when considering the 4-CFR, they were body weight, estimated time from trauma to the therapeutic intervention, spine involvement, hospitalization time and the TTRS. 

Considering the epidemiology within this unowned cat population, intact male cats (46%) were over-represented, as previously reported in the traumatized feline patient [6,22,23,24]. This finding might be justified by the behavioral habits of these cats; indeed, compared to the other sex categories, intact males are less likely to settle down in a defined territory and tend to roam in search of resources or females for mating [25]. These movements expose them to the possibility of being traumatized by road traffic accidents or to bite wounds injuries [6,9,26].

In our study, we found a significant association between the animal’s body weight and the outcome considering the 4-CFR as previously reported for feline patients affected by bite wounds [27]. The small size of the feline patients predisposed them to multiple and more severe injuries following a traumatic event [28]. On the contrary, heavier cats were less likely to survive from high-rise syndrome in a recent study [29]. Probably, the forces generated by the traumatic impact from road traffic accidents are distributed over the body surface and larger animals can absorb and dissipate forces more than smaller ones [28], whereas heavier cats falling from a height produces a stronger impact force when they reach the landing surface [29]. Hence, our results reinforce the hypothesis that smaller and lighter subjects suffer from more severe traumatic injuries, which are more complex to manage and jeopardize the possibility of restoring an acceptable function of the affected organs. On the other hand, it might not be excluded that heavier cats might not be referred due to more severe lesions leading to death, possibly linked to less agility [29].

In our cohort, unowned cats undergoing surgery within 7 days after trauma showed a 4.6 times less chance of reaching CFR considering the 4-CFR. Even though the impact of surgical timing on the CFR was not evaluated as such, it is possible that longer surgical timing should be applied in polytraumatized cats. Studies debating over timing of surgical intervention in traumatized cats with diaphragmatic hernia showed no effect of timing on complications and outcome [30,31]. On the other hand, surgical timing is a turning point in the management of polytraumatized human patients. Indeed, it has been discussed if a delayed or early definitive surgical treatment is the best approach, with both bearing pros and cons [32]. Early definitive surgery has been gradually replaced with damage control surgery, aimed at controlling bleeding and contamination [32,33]. The optimal timing of surgical intervention for orthopedic procedures among small animal patients suffering trauma is currently unknown. Primary insult, surgical burden, i.e., operative intervention and second hit phenomena, and the response of the immune system occur in polytraumatized patients. Nowadays, these conditions are considered critical factors directly affecting the clinical outcome in humans and should also be considered important for small animal patients [34]. 

Variables significantly associated with the CFR, but without any significance at the post hoc analysis were BCS and the skeletal radiographic score considering the 2-CFR, and lesions involving the spine considering the 4-CFR. It may be hypothesized that the lack of association between the BCS and the CFR is linked to the subjectivity of this score, as previously discussed by other authors [29]. Moreover, the number of classes represented among the nine levels of the BCS might be responsible for some data scattering, with each class including not enough subjects to reach significance at the post hoc analysis.

The skeletal score was associated with the CFR at the univariate and multivariable analysis, but no association could be detected at the post hoc evaluation. As expected, the skeletal score showed an association with a better outcome in terms of improved mobility. The thoracic score applied in our study was modified from a previously published radiographic score [18], which showed a similar lack of association with outcome. The TTRS was available for a smaller subgroup within our sample. Nonetheless, it resulted as an independent predictor of the CFR. The radiographic examination represents a widely accessible diagnostic tool in the emergency setting and bears the pros of being cheaper compared to other advanced diagnostic imaging modalities, even though computed tomography has been demonstrated to be more accurate for the evaluation of polytraumatized cats [35]. The skeletal, thoracic, abdominal, and total radiographic scores were proposed to quantify and summarize the radiographic abnormalities identifiable in the feline patients included, and they were not intended to simplify the complexity of lesions that may occur in polytraumatized patients. Nonetheless, they might be applied in the clinical setting to help the clinicians in estimating the return of the patient to a good quality of life. 

Lesions of the spine were recorded in 15% of the patients, similar to what was previously reported [8]. Spine involvement has been shown as a negative prognostic factor on survival after trauma [9,36]. Spine involvement did not reach significance at the post hoc statistical evaluation. It is important to underline that, in the present study, cats with sacroiliac fracture-luxation were included in the spinal involvement category. It might be hypothesized that the relatively low number of patients experiencing spine involvement in our sample, as well as the inclusion of the less severe aforementioned lesions, has influenced the marginal association with CFR. 

The results from the 30- and 60-day clinical re-evaluations, graded as positive or negative according to the clinical improvement of the affected regions, were significantly associated with the 2-CFR. These variables should not be regarded as influencing the outcome, but rather as predictive of it. Indeed, according to our results, a negative score at the 30- and 60-day clinical re-evaluations should alert the clinician to further deepen the patient’s assessment, such as with additional examinations, continued diagnostics, or clinical-metabolic monitoring. On the other hand, in our experience, the absence of post-therapy complications led to an uneventful recovery and thus to a higher probability of a good return to function, even in this peculiar feline population.

The definition of CFR proposed by the authors was conceived according to a previous orthopedic subjective clinical outcome evaluation [15]. This latter clinical outcome was defined as full, acceptable, or unacceptable [15]. Starting from this idea, in order to classify our therapeutic results, we developed a two-category subjective clinical outcome, CFR vs. no-CFR, with the intent to easily classify traumatized unowned cats bearing orthopedic injuries as with or without permanent disabilities, i.e., the patient could—or could not—return to free-roaming life. The time-point for the CFR evaluation was set at 90 days post-treatment to balance the need to give enough time for recovery from traumatic orthopedics injuries, e.g., bone fractures, and the need to have a reliable evaluation for the final arrangement decision making. It has been reported that the time to function could vary from 21 to 182 days with a median time from 42 to 53 days in cats who sustained a humeral fracture [37]. Thus, the 90-day evaluation seems to be a good compromise to appreciate the impact of the therapy on traumatized cats.

The overall mortality in this study was 17%, but no patient died from injuries directly related to the trauma, in contrast with other studies [9,36], even if in our peculiar sample of unowned cats, the most severe cases might be hypothesized to die before referral. Comorbidities, which showed a subtle association with the lack of functional recovery only at the univariate analysis considering the 2-CFR, were detected in 31% of the patients in our sample and determined the death of three of them. Such a fatal outcome, recorded in 6% of the cats included, might explain the association of comorbidities with no-CFR. The other five cats died from panleukopenia; cats dying from panleukopenia were younger compared to all other outcome classes, and lighter than cats experiencing CFR considering the 4-CFR. Indeed, a higher risk of dying has been reported in sheltered kittens with lighter body weight, with low BCS, and those diagnosed with panleukopenia [17]. Moreover, it has been previously reported that panleukopenia is a common cause of mortality in kittens and young cats [6], both among unowned and owned feline patients [38]. Whereas a multi-patient environment, such as during hospitalization or sheltering, and an incomplete or absent vaccination protocol has been demonstrated a predisposing factor for the development of panleukopenia [39]; we supposed that death of these patients from feline panleukopenia should not be considered entirely independent from the traumatic event. Indeed, it has been widely demonstrated that a series of biochemical events take place following trauma in human patients, which lead to a state of uncontrolled inflammation and immunosuppression that continues for some time after the trauma [40]. Furthermore, because of this condition, the patient bears a higher risk for the systemic inflammatory response syndrome, accompanied by the compensatory anti-inflammatory response syndrome and a subsequent dysfunction of the immune system, which is unable to protect the subject from possible secondary infections [41,42]. 

Our study inherits some limitations. First, due to its retrospective nature, we could not apply previously validated scales, such as the animal trauma triage score (ATTs) or the modified Glasgow coma scale (mGCS) [43,44], since data records did not always include all the variables requested for their computation. Few studies calculated the ATTs from retrospectively collected data [10,26], and it resulted with an association with the need for transfusion and hospital discharge in cats with pelvic injury [10], whereas most research evaluated retrospectively previously recorded ATTs [9,27,43,45]. On the other hand, the mGCS had been only evaluated retrospectively from data collected on arrival [27,36,45,46]. It would be interesting to explore the performance of our TTRS in comparison to the other already validated scales, and eventually identify their predictive cut-offs on the CFR. 

It is possible that some cats classified as no-CFR considering both the 2-CFR and 4-CFR would have had further improvement of the clinical symptomatology leading to a CFR with a follow-up longer than 90 days. Moreover, when considering the 2-CFR, patients who died within 90 days post-therapy were classified as no-CFR. It has been reported that approximately 75% of feline deaths related to trauma, with the exclusion of euthanasia for economic reasons, occurs in the first 24 h post-trauma [36]. In our population, the median estimated time from trauma to referral was 3 days, which seems longer than the prompt referral of owned cats. In these circumstances, patients are first evaluated by the public veterinary service, and they are referred to the VTHUM thereafter. Only in a minority of cases, citizens carry the traumatized unowned cats immediately to VTHUM. It might be hypothesized that the most severe cases died before referral to our facility or were not discovered at all. On the other hand, it might be possible that only severely affected unowned cats were referred, whereas those bearing slight injuries might be directly treated or not recognized. Hence, our sample should not be considered representative of all traumatized unowned cats. 

Estimation of age might also be seen as a limitation. Indeed, after completion of permanent dentition [47] it is hard to objectively and precisely estimate age based on dentition. In such cases, information from the caregivers and the veterinary public service, when available, represent the only source available.

In Italy the economic burden of the management of unowned cats is supported by the public health system and to a lesser extent by non-profit associations. It would be interesting to obtain precise data about the veterinary expenses, i.e., diagnostic, treatment, and hospitalization costs, of traumatized unowned cats and whether the CFR differs in terms of cost from the no-CFR. Eventually, the cheaper practices not influencing the CFR might be chosen, with equal effectiveness. In this perspective, our analysis was not able to detect any significant difference between the use of internal vs. external fixation, and most of the acute orthopedic lesions were treated by internal fixation. This choice was dictated by the consideration that internal fixation needs less post-operative care compared to external fixation. Even though internal fixation has a higher initial cost, the overall expense should be evaluated taking into account the post operative management, but further data should be collected to answer this point. Finally, the lack of association between 2-CFR and adoption rate, sheltering, and return to their original location in a feline group could have been influenced by the low number of patients in the no-CFR class once dead patients were removed. Even if non-significant, a larger proportion of no-CFR patients were sheltered, compared to unowned cats experiencing CFR, and none of them were returned to their original location. A larger sample is needed to properly evaluate the real lack of association between CFR and all non-significant variables. The relative low number of patients included in the present study might have caused a lack of statistical power, leading to non-significant results.

## 5. Conclusions

The management of traumatized unowned cats with orthopedic injuries lead to a good percentage of CFR. The timing for therapeutic intervention should be tailored to the specific case, considering that stabilization procedures and clinical classification are of paramount importance in the management of traumatized patients. Radiographic evaluation should always be considered, possibly including the whole patient for a complete radiographic assessment, and not only the most clearly traumatized region. Traumatized unowned kittens have an increased risk of death from panleukopenia infection, possibly due to immunodepression secondary to the traumatic event. Moreover, the presence of comorbidities, even if not influencing the possibility of a CFR, might lead to the death of the patients. Hence, caregivers and veterinarians should be warned that younger and lighter cats, the presence of comorbidities, multiple lesions and spine involvement may all lead to a guarded to poor prognosis. 

## Figures and Tables

**Figure 1 vetsci-11-00059-f001:**
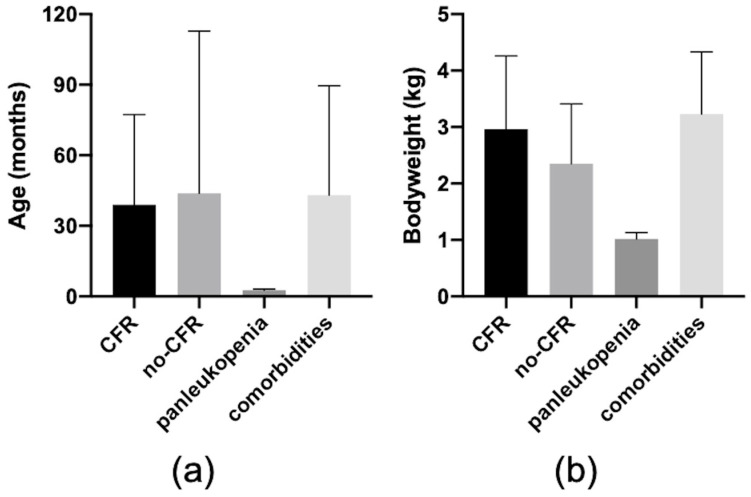
Graphic representation of the mean ± standard deviation of the variables significantly associated with death from panleukopenia. In (**a**), age in months in the four classes (CFR vs. no-CFR vs. death from panleukopenia vs. death from comorbidities). In (**b**), body weight in kilograms in the four classes (CFR vs. no-CFR vs. death from panleukopenia vs. death from comorbidities).

**Table 1 vetsci-11-00059-t001:** Variables significantly associated with the 2-CFR categories (CFR vs. no-CFR).

Variable	CFR (N, %)	No-CFR (N, %)	Univariate (*p*-Value)	Youden’s *J*	Fisher’s *F* (*p*-Value, O.R., 95%CI)	Multivariable (*p*-Value)
Body weight			0.027	0.36	0.047; 4.7; 0.9–24.5	
≤3.3 kg	19 (40%)	12 (25%)				
≥3.4 kg	15 (31%)	2 (4%)				
BCS			0.043		0.16 n.s.	0.004
TTI			0.054	0.36	0.030; 4.6; 1.1–19.7	
≤7 days	15 (31%)	11 (23%)				
≥8 days	19 (40%)	3 (6%)				
Spine			0.09		0.08 n.s.	
Yes	3 (6%)	4 (8%)				
No	31 (65%)	10 (21%)				
Comorbid.			0.08		0.09 n.s.	
Yes	8 (17%)	7 (14.5%)				
No	26 (54%)	7 (14.5%)				
Skeletal s. ***			0.055	0.28	0.018; 0.07; 0.01–0.71	<0.0001
≤2	32 (67%)	9 (19%)				
=3	1 (2%)	4 (8%)				
TTRS ***			0.02	0.40	0.11 n.s.	0.008
≤1	13 (27%)	2 (4%)				
≥2	8 (17%)	7 (15%)				
30 days ***			0.02		0.03; 12; 1.5–95.0	
positive	32 (80%)	2 (5%)				
negative	2 (5%)	4 (10%)				
60 days ***			0.003		0.008; 33; 2.6–423.0	
positive	33 (82.5%)	2 (5%)				
negative	1 (2.5%)	4 (10%)				
Outcome	34 (71%)	14 (29%)*^*^*				
Adopted ^‡^	22 (46.2%)	4 (8.4%)				
Sheltered ^‡^	3 (6.2%)	2 (4%)				
Returned ^‡^	9 (19%)	0 (0%)				

CFR: complete functional recovery; O.R.: odds ratio; age: estimated age; BCS: 9-points scale body condition score [16]; n.s.: non-significant; TTI: time to therapeutic intervention; Comorb.: presence of comorbidities; Skeletal s.: skeletal score—*: 2 missing values, 1 from the CFR group (2%) and 1 from the no-CFR group (2%); TTRS: total trauma radiographic score—*: 18 missing values, 13 (27%) from the CFR group, 2 (4%) from the no-CFR group, 1 (2%) from the death from comorbidities group, 2 (4%) from the death from panleukemia group. The 30 days and 60 days: re-evaluations—*: 8 missing values, 3 (6.2%) from the death from comorbidities group, 5 (10%) from the death from panleukemia group. Outcome *: includes the 8 patients who died prior to the 30-day re-evaluation. ‡: these three rows should be summed, plus the who 8 patients died prior to the 30-day re-evaluation.

**Table 2 vetsci-11-00059-t002:** Variables non-significantly associated with CFR. On the left, 2-CFR; on the right, 4-CFR.

CFR vs. No-CFR		CFR vs. No-CFR vs. Comorbidities vs. Panleukemia	
Variable	*p* Value	Variable	*p* Value
Estimated age	0.43	Estimated age	*sig.*
Sex	0.31	Sex	0.20
BCS	*sig.*	BCS	0.13
Estimated time from trauma to referral	0.21	Estimated time from trauma to referral	*sig.*
Type of acute orthopedic lesions	0.16	Type of acute orthopedic lesions	0.27
Presence of chronic orthopedic lesions	0.11	Presence of chronic orthopedic lesions	0.32
Single or multiple traumatic lesions	0.65	Single or multiple traumatic lesions	0.93
Interaction of acute and chronic orthopedic lesions	0.30	Interaction of acute and chronic orthopedic lesions	0.37
Involvement of head	0.57	Involvement of head	0.88
Involvement of forelimb	0.97	Involvement of forelimb	0.19
Involvement of hindlimb	0.79	Involvement of hindlimb	0.57
Involvement of pelvis	0.52	Involvement of pelvis	0.87
Involvement of thorax	0.36	Involvement of thorax	0.13
Involvement of abdomen	0.17	Involvement of abdomen	0.44
Interaction of involved regions	0.87	Interaction of involved regions	0.75
Concomitant soft tissues or neurological traumatic lesions	0.31	Concomitant soft tissues or neurological traumatic lesions	0.30
Blood transfusion post-therapy	0.11	Blood transfusion post-therapy	0.21
Esophagostomy feeding tube	0.77	Esophagostomy feeding tube	0.49
Type of therapeutic intervention	0.28	Type of therapeutic intervention	0.68
Internal or External fixation	0.81	Internal or External fixation	0.79
Post-therapy complications	0.97	Post-therapy complications	0.52
Hospitalization time	0.12	Hospitalization time	0.23
Skeletal radiographic score	*sig.*	Skeletal radiographic score	0.52
Thoracic radiographic score	0.82	Thoracic radiographic score	0.36
Abdominal radiographic score	0.43	Abdominal radiographic score	0.53

CFR: complete functional recovery; *sig.*: statistically significant, refer to Table 1; BCS: 9-points scale body condition score [16].

**Table 3 vetsci-11-00059-t003:** Variables significantly associated with 4-CFR categories (CFR vs. no-CFR vs. death from comorbidities vs. death from panleukemia).

Variable	CFR (N, %)	No-CFR (N, %)	Comorbidities (N, %)	Panleukopenia (N, %)	Univariate (*p*-Value)	Multivariable (*p*-Value)
Age					0.07	
Body weight					0.003	<0.0001
≤3.3 kg	19 (40%)	6 (13%)	1 (2%)	5 (10%)		
≥3.4 kg	15 (31%)	0 (0%)	2 (4%)	0 (0%)		
TTI					0.10	<0.0001
≤7 days	15 (31%)	4 (8.4%)	3 (6.2%)	4 (8.4%)		
≥8 days	19 (40%)	2 (4%)	0 (0%)	1 (2%)		
Spine					0.09	<0.0001
Yes	3 (6.2%)	3 (6.2%)	0 (0%)	1 (2%)		
No	31 (64.8%)	3 (6.2%)	3 (6.2%)	4 (8.4%)		
Hospital.					0.014	<0.0001
Comorbid.					0.03	
Yes	8 (17%)	3 (6.2%)	3 (6.2%)	1 (2%)		
No	26 (54%)	3 (6.2%)	0 (0%)	4 (8.4%)		
TTRS *					0.02	<0.0001
≤1	13 (27.3%)	1 (2%)	1 (2%)	0 (0%)		
≥2	8 (17%)	3 (6.2%)	1 (2%)	3 (6.2%)		
30 days *					0.006	
positive	32 (67.4%)	2 (4%)	0 (0%)	0 (0%)		
negative	2 (4%)	4 (8.4%)	0 (0%)	0 (0%)		
60 days *					0.004	
positive	33 (69.4%)	2 (4%)	0 (0%)	0 (0%)		
negative	1 (2%)	4 (8.4%)	0 (0%)	0 (0%)		
Outcome	34 (71%)	6 (29%)	3	5		

CFR: complete functional recovery; O.R.: odds ratio; age: estimated age; n.s.: non-significant; TTI: time to therapeutic intervention; hospital.: hospitalization length; comorb.: presence of comorbidities; TTRS: total trauma radiographic score—*: 18 missing values, 13 (27.3%) from the CFR group, 2 (4%) from the no-CFR group, 1 (2%) from the death from comorbidities group, and 2 (4%) from the death from panleukemia group. The 30 days and 60 days: re-evaluations—*: 8 missing values, 3 (6.2%) from the death from comorbidities group, 5 (10%) from the death from panleukemia group. Outcome *: includes the 8 patients died prior to the 30-day re-evaluation.

## Data Availability

Most of the raw data used in this study have already been reported in the Appendix A. All data are available from the corresponding author upon reasonable request.

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
