# Peer review of "Potential Factors Influencing Complete Functional Recovery in Traumatized Unowned Cats with Orthopedic Lesions—A Cohort Study"

_vetsci, 2024, doi:10.3390/vetsci11020059_

Round 1

Reviewer 1 Report

Comments and Suggestions for Authors

This is an incredibly valuable study as no one has published on this topic. And very few locations are likely to have enough cases from unowned free-roaming cats to be able to do this type of analysis. I do have many comments and questions below—I’m extremely excited to see the revision! Overall, one set of comments are about terminology, which is complicated even for native English speakers if they don’t work in the area of free-roaming cats. Another major set is about statistical analyses. Not that there is a problem, but that it is still quite confusing about what was done and found. I’ve made a number of specific recommendations which I hope the authors will find useful. The final group of comments is about requesting additional information either in a different place in the manuscript or added to the manuscript.

I understand that the authors were interested in a primary outcome of return to function or not. Hence the use of the 2 category CFR. However, the discovery of the panleukopenia and co-morbidity causes of death in this population of cats appears to be vitally important. And understanding how those 2 subgroups differ from the cats without CFR but no other health issues also seems critical to help readers make decisions about which cats are likely to do well (most of them), what to advise the caregivers, and where to spend the limited resources. I’m therefore wondering if the manuscript should ONLY use the 4 category CFR for all the univariable and multivariable analyses.

Use past tense in the manuscript please.

Simple Summary: There really isn’t any way to identify stray from less socialized or born unowned or feral cats. If the cat is an owned outdoor cat, the cat could have similar risks of trauma to unowned cats. I suggest using the term “unowned free-roaming cat” (frc) throughout. That is what these cats are. Additionally, most are moving away from the term “colony” to use “group” instead for a variety of reasons. Please edit throughout.

Line 18-19: were all cases “referred” in the sense that they were seen in private veterinary practice and recommended to go to the teaching hospital? We don’t find this out until lines 447-9. Or were they simply told by various people that they could take the cat to the teaching hospital? Who brings the stray cats into the hospital? Please clarify in introduction. Please include in the methods section for clarification. And that some DO come directly to the hospital. Which cats were those (perhaps note in one of the supplement tables)?

Line 21: What is complete functional recovery? Does a 3-legged cat have that? Perhaps instead of using this term here, use a general description of what it means: able to move well and perform normal maintenance behaviors.

Line 23: “…lighter cats who survive to presentation experience”. Add bolded words for context here.

Abstract Line 31: I think that “reintroduction” (and on line 67 “release”) isn’t really what is meant…I believe that the authors mean return to their original location? At least I hope so. Please edit throughout. And for adoption, many sorts of limitations could be feasible depending on the adopter and lifestyle of the adopted cat. Same comment for lines 76-7.

Line 34: retrospectively collected…add “from the medical records”.

Line 38: please use panleukopenia here and throughout the manuscript.

Line 40: please include the % who returned to CFR.

Line 41: I expect it is also the young age (as the authors discuss towards the end of the manuscript) that causes the problems with panleukopenia as they have either not been exposed or vaccinated yet and have been stressed by trauma (lines 298-300 would support this idea…or at least that weight is a stronger predictor than age. And since weight is also easier to obtain accurately this is a good thing). Also include that longer time likely meant that the trauma was less severe overall, and therefore, cats who were severely traumatized and didn’t come immediately to the hospital didn’t live long enough to be seen and treated. Many people only read the summary or abstract so be sure to consider what has to be included.

Line 58: free-roaming and owned aren’t mutually exclusive. I understand the authors are using the references terminology, but that reference is most likely using “stray” cats to be unowned as compared to owned cats brought in for care by owners. So, using owned and unowned will be a better way to say this. There is an assumption that stray cats are more likely to be free-roaming but in the UK most owned cats also are allowed outside. I suspect the finding is due to owner observation of illness vs unowned cats being found by roads with trauma, but the referenced abstract isn’t clear. So, some additional context for this statement based on reference 7 would be helpful.

Line 68: how is adoption different from sheltering? Does this mean that cats were already adopted by the end of the study time compared to cats who were still in the shelter awaiting adoption? Please clarify. Did the hospital work with many different shelters or only one or two? Because adoption is a function of how the shelter operates, the number of different shelters is important to include.

Line 69: please clarify what CFR means in this context; I see it on line 120 but would like it to be included here as well. What about the cat’s ability to posture for urination or defecation? That would seem an important criterion. Also, lines 124-7. Please move all the information about how CFR is defined into one location fairly early in the manuscript. Is there a reference for this definition or is this being created by the authors? Please clarify in the text.

Line 76: is this statement about adoption for Italy? In general? Please reference or indicate that this is the authors’ opinions.

Line 89: how was time from trauma estimated? Because this variable is important, any insight into how this information was obtained from the caregiver and recorded in the medical record is helpful.

Line 91 & 108: both list collecting data on the clinical re-evaluation at the 3 time points.

Line 92: Exclusion would also include cats without all of the data included in the medical record. Are these data points listed above typically included in the records? Please add to text for clarity as lack of these data points can result in cats being excluded.

Line 94-5: were all cats routinely testing for these viruses? Please include in text.

Line 96: how was age estimated? Please include this in the text.

Line 97: please include which BCS was used in the text (at least that it was the 9 point one).

Line 110-11: so, these evaluations were just scored as improved or not? I don’t see any descriptive data about these 3 time periods in the results. Please add.

Line 140-2: is this first score not adapted from other work? Please clarify in the text. Same question for the abdominal score.

Lines 161-2: this is how a variable is defined and should be included above where the variables are listed (and defined as needed).

Line 162: what analyses were used? Are these what is listed in lines 169-70? Please order this whole section by descriptive statistics, univariable statistics, multivariable statistical analyses and for the latter two include the list of independent variables and dependent variables. How were the variables included in the model—all put in and then just reporting on which were significant still? Please add to text. How well did the logistic models fit? Please include some information about how this was evaluated here and, in the results, add what was found. Why wasn’t a multinomial logistic regression used for the 4-category outcome variable?

Line 185: were any excluded due to missing data in the medical records? Thank you for including this information for context during the time period!

Line 189-194: it would be helpful to either provide a summary by category of CFR for these 4 variables or add a mean/median/percentage as appropriate to the supplemental table.

Line 202: please add that 3 cats (I think) had both acute and chronic orthopedic lesions for clear context here.

Line 230: is this cat with immediate surgery one of the 2 without imaging? Please phrase more clearly (perhaps: Of these 2 patients without imaging,).

I like Table 1 and would like the variables to be aligned so that it is easy to tell which were not significant for both approaches to classification.

I would then like an additional table for the univariable and multivariable analyses which included any variable significantly associated with CFR so that much of the text on pages 6-8 would be in a table instead. It might look something like below (order of columns should be consistent throughout with order…seems to be different in supplement than in text?):

Note that I just roughly calculated the numbers and % from the supplementary tables. With these data, I get a p-value of 0.2 for the chi-square. Please double check the data and analyses.

Variable

CFR (N, row %)

No-CFR (N, %)

Panleukopenia

(N, %)

Other Death (N, %)

Univariable p-value

OR & 95% CI

Logistic regression p-value

OR and 95% CI

Body weight

 <=3.3 kg

20, 63%

5, 16%

5, 16%

2, 6%

Baseline category

--

--

--

>=3.4 kg

14, 88%

1, 6%

0, 0

1, 6%

0.047

0.9-25

0.16

n/a

Outcome

0.13

--

--

--

 Adopted*

 Sheltered

Returned

*and I’m not sure I can get the data for these 12 cells from the manuscript. Only the CFR yes/no is included.

Line 235: Please include descriptive data on the length of time to CFR…that will help others understand how long this might realistically take to reach.

Line 260: is this the final CFR assessment? Are the lines 284-5 data all information from whatever the final time point was? Is there a reason why this was not examined in a multivariable analysis?

Line 261: what post hoc analysis? The multivariable logistic model?

Line 273: these two re-evaluations are nested within individual cats—if they are both significant then they aren’t strongly associated with each other. However, treating them as independent variables in the multivariable analysis (I’m assuming that is what this is?) when they aren’t really, should be addressed in the discussion.

Line 280-1: this null result could just be included as a comment in the text as it is now.

Supplemental tables: please make sure that the titles are stand alone and explain the context of the study and all abbreviations. For Table 1: the Time to referral is not the same as Time to treatment so please be clear how that time was defined. Similarly for Table 2 for Time to therapy and Hospitalization time (is this after therapy to outcome or disposition, I’m assuming). What does “recurrence” mean in as a complication—I’m guessing of the luxation? Please clarify in table. Table 3: these are scores from 0-10, aren’t they? So why are these all words? I think the text is useful so perhaps just change the column headings and add a total trauma radiographic score column for the actual scoring. I also don’t see a summary of outcome/disposition anywhere (sheltered, adopted, euthanized/died) so perhaps add a column to Table 1 in the supplement with what happened to each cat.

Line 291: what post hoc analyses were performed? Is this simply the results of the Wald tests for logistic regression? Please clarify.

Line 316-7: that isn’t what is stated in lines 298-300. I think determining which set of CFR analyses are most important and being clear throughout the manuscript which CRF classification was used will be critical.

The discussion feels a little mixed up. It feels like the paragraphs starting on line 349 is quite important for this study—the authors are seeking predictors of survival from trauma. Or perhaps the more important finding is that surgery isn’t needed all the time? Please consider the fundamental purpose of this manuscript and organize the discussion from most to least important.

The first paragraph does give a high-level statement of the findings but uses variables that aren’t clearly defined. Please make this more stand alone and describe what are the 30- and 60-day results (improvement in function at those time points post-treatment). Lines 278-280 list a different set of significant variables.

Line 318: this should have been described in the methods or at least referenced there. And permanent disabilities like losing a leg are generally not a limitation in returning a cat to a relatively safe outdoor location. Please clarify in the text.

Lines 324-5: here is where having the data from the actual mean/median time to CFR in the paper would be helpful. The authors could indicate if their estimate was accurate and if not, what would be a better estimate.

Line 340: this could also be selection bias where heavier cats didn’t survive (maybe they were slower) to be referred to the hospital with more severe injuries (I doubt this is true, but it is another explanation for the findings which should be mentioned).

Paragraph starting on line 371: it is also possible that BCS was less strongly associated with CFR than weight.

Line 383-4: post hoc analysis wasn’t described in the methods please add it there. And this categorization wasn’t mentioned in methods either.

Line 389: TTRS was associated according to line 299.

Line 404: likely because stray cats died prior to being referred in…assuming that these references apply to owned cats.

Line 440: but confounding in this study should be able to be assessed and evaluated, shouldn’t it?

Conclusions: many readers only look at the conclusion. Please include a bit more detail about the findings and any implications from the results about whether some cats should be treated and some euthanized (doesn’t look that way but important to state) as well as how to counsel caregivers or other veterinarians about what to expect.

Comments on the Quality of English Language

In a few places the language choice makes understanding the point difficult. In most places the language doesn't get in the way of understanding. However, it would be easier to make the important points of the study with some English editing.

Author Response

Comments and Suggestions for Authors

This is an incredibly valuable study as no one has published on this topic. And very few locations are likely to have enough cases from unowned free-roaming cats to be able to do this type of analysis. I do have many comments and questions below—I’m extremely excited to see the revision! Overall, one set of comments are about terminology, which is complicated even for native English speakers if they don’t work in the area of free-roaming cats. Another major set is about statistical analyses. Not that there is a problem, but that it is still quite confusing about what was done and found. I’ve made a number of specific recommendations which I hope the authors will find useful. The final group of comments is about requesting additional information either in a different place in the manuscript or added to the manuscript.

Authors: we appreciated the depth at which the Reviewer evaluated our work, highlighting both the strengths and weaknesses. We are grateful for the job made by the Reviewer in improving our manuscript.

I understand that the authors were interested in a primary outcome of return to function or not. Hence the use of the 2 category CFR. However, the discovery of the panleukopenia and co-morbidity causes of death in this population of cats appears to be vitally important. And understanding how those 2 subgroups differ from the cats without CFR but no other health issues also seems critical to help readers make decisions about which cats are likely to do well (most of them), what to advise the caregivers, and where to spend the limited resources. I’m therefore wondering if the manuscript should ONLY use the 4 category CFR for all the univariable and multivariable analyses.

Authors: we thank the reviewer for the suggestion, we thought about retaining only the statistical analysis for the 4 classes, i.e., CFR, no-CFR, death for panleukemia, death for comorbidities, but according to the further comments of the Reviewer, we think that the information obtained from the 2 category CFR (CFR vs. no-CFR) statistics have been considered also of great value.

Use past tense in the manuscript please.

Authors: we thank the reviewer for the suggestion, we changed to past tense whenever referring to our or other studies’ results.

Simple Summary: There really isn’t any way to identify stray from less socialized or born unowned or feral cats. If the cat is an owned outdoor cat, the cat could have similar risks of trauma to unowned cats. I suggest using the term “unowned free-roaming cat” (frc) throughout. That is what these cats are. Additionally, most are moving away from the term “colony” to use “group” instead for a variety of reasons. Please edit throughout.

Authors: we changed stray and free-roaming in unowned throughout. We changed colony in group.

Line 18-19: were all cases “referred” in the sense that they were seen in private veterinary practice and recommended to go to the teaching hospital? We don’t find this out until lines 447-9. Or were they simply told by various people that they could take the cat to the teaching hospital? Who brings the stray cats into the hospital? Please clarify in introduction. Please include in the methods section for clarification. And that some DO come directly to the hospital. Which cats were those (perhaps note in one of the supplement tables)?

Authors: we thank the reviewer for the request of clarification. Actually, as explained in the discussion, as underlined by the reviewer, we work as a second practice in cooperation with the veterinary public service. Only in very few cases citizens spontaneously and directly bring unowned animals to the veterinary teaching hospital. Even less, private practices are involved only for the very first aid and stabilization of the most severe cases, and then they are referred to the veterinary public service.

To better reply to the Reviewer#1 we performed an adjunctive check of our medical record that confirmed that we can say that none of the cases reported in our study were referred by citizens, nor from private practices. All patients were referred by the veterinary public service, and we added in the simple summary (line 21 _REV1, line 20 _amended), in the abstract (line 37 _REV1, line 34 _amended), in the introduction (line 66 _REV1, line 59 _amended), and in the materials section (line 104 _REV1, lines 96-97 _amended).

Line 21: What is complete functional recovery? Does a 3-legged cat have that? Perhaps instead of using this term here, use a general description of what it means: able to move well and perform normal maintenance behaviors.

Authors: we thank the reviewer for the suggestion, we rephrased accordingly (lines 18-19 _REV1, lines 18-19 _amended)

Line 23: “…lighter cats who survive to presentation experience”. Add bolded words for context here.

Authors: we thank the reviewer for the suggestion, we rephrased accordingly (line 25 _REV1, line 24 _amended).

Abstract Line 31: I think that “reintroduction” (and on line 67 “release”) isn’t really what is meant…I believe that the authors mean return to their original location? At least I hope so. Please edit throughout. And for adoption, many sorts of limitations could be feasible depending on the adopter and lifestyle of the adopted cat. Same comment for lines 76-7.

Authors: we thank the reviewer for the suggestion, we changed both reintroduction and release with “return to their original location”. We added a comment in the introduction about the adoption, hoping we understood correctly the Reviewer’s intention, but we do not have enough in the abstract. “Adoption might be influenced by several factors, such as location, characteristic of the cat, and previous humane interactions [Kilgour], and it is felt more unlikely for cats needing continuous management or medical care.” (lines 94-96 _REV1, lines 86-88 _amended)

Line 34: retrospectively collected…add “from the medical records”.

Authors: changed as suggested. (line 38 _REV1, line 35 _amended)

Line 38: please use panleukopenia here and throughout the manuscript.

Authors: changed as suggested throughout the manuscript.

Line 40: please include the % who returned to CFR.

Authors: changed as suggested. (line 41 _REV1, line 37 _amended)

Line 41: I expect it is also the young age (as the authors discuss towards the end of the manuscript) that causes the problems with panleukopenia as they have either not been exposed or vaccinated yet and have been stressed by trauma (lines 298-300 would support this idea…or at least that weight is a stronger predictor than age. And since weight is also easier to obtain accurately this is a good thing). Also include that longer time likely meant that the trauma was less severe overall, and therefore, cats who were severely traumatized and didn’t come immediately to the hospital didn’t live long enough to be seen and treated. Many people only read the summary or abstract so be sure to consider what has to be included.

Authors: we thank the reviewer for the suggestion. We tried to change the abstract following all the interesting ideas, underlying that both estimated age and bodyweight might have an influence, particularly on the panleukemia. Moreover, we added a phrase about time to intervention, as for the Reviewer #1 advice.

Line 58: free-roaming and owned aren’t mutually exclusive. I understand the authors are using the references terminology, but that reference is most likely using “stray” cats to be unowned as compared to owned cats brought in for care by owners. So, using owned and unowned will be a better way to say this. There is an assumption that stray cats are more likely to be free-roaming but in the UK most owned cats also are allowed outside. I suspect the finding is due to owner observation of illness vs unowned cats being found by roads with trauma, but the referenced abstract isn’t clear. So, some additional context for this statement based on reference 7 would be helpful.

Authors: as for previous comment of the Rewiever#1, we changed to “unowned” throughout the manuscript. We tried to clarify the findings of ref. 7 as suggested (lines 71-73 _REV1, lines 64-66 _amended)

Line 68: how is adoption different from sheltering? Does this mean that cats were already adopted by the end of the study time compared to cats who were still in the shelter awaiting adoption? Please clarify.

Authors: if we understood correctly what the Reviewer#1 is asking, we just want to clarify in this introduction section that what was reported is a general explanation of the possible arrangement of an unowned cat after their medical management. This section does not refer to the actual destiny of the cats included in our study.

Did the hospital work with many different shelters or only one or two? Because adoption is a function of how the shelter operates, the number of different shelters is important to include.

Authors: we operate at least with 6 different shelters. Anyhow, we did not include this data since all the shelters with which we cooperate are directed by the same and unique veterinary public service, hence management, adoption and medical procedures are centralized and shared among such shelters.

Line 69: please clarify what CFR means in this context; I see it on line 120 but would like it to be included here as well. What about the cat’s ability to posture for urination or defecation? That would seem an important criterion. Also, lines 124-7. Please move all the information about how CFR is defined into one location fairly early in the manuscript. Is there a reference for this definition or is this being created by the authors? Please clarify in the text.

Authors: we thank the reviewer for the suggestion. We clarified in the introduction how the CFR has been defined generally. We also moved at the second paragraph of the materials section the definition of the CFR, including that it was modified from a previous published scale for orthopedic outcomes. We also underline that in addition to the straining also getting the posture for urinating and defecating was an important point, which we already included in the “straining”. (lines 81-83 _REV1, lines 74-76 _amended)

Line 76: is this statement about adoption for Italy? In general? Please reference or indicate that this is the authors’ opinions.

Authors: in Italy there is actually a strong problem with adoption in general, and it is quite the rule that possible owners refuse to adopt pets with chronic medical issues. Moreover, these patients might not be easily manageable by all potential owners (e.g., urinary and fecal retention, urinary or fecal incontinence, etc.). Anyhow, we did include some references which sustain similar link between medical issues and adoption (ref 12-14).

Kilgour, R.J.; Flockhart, D.T.T. Direct and Indirect Factors Influencing Cat Outcomes at an Animal Shelter. Front Vet Sci 2022, 9, 766312, doi:10.3389/fvets.2022.766312.

Murphy, M.D.; Larson, J.; Tyler, A.; Kvam, V.; Frank, K.; Eia, C.; Bickett-Weddle, D.; Flaming, K.; Baldwin, C.J.; Petersen, C.A. Assessment of owner willingness to treat or manage diseases of dogs and cats as a guide to shelter animal adoptability. J Am Vet Med Assoc 2013, 242, 46-53, doi:10.2460/javma.242.1.46.

Lepper, M.; Kass, P.H.; Hart, L.A. Prediction of adoption versus euthanasia among dogs and cats in a California animal shelter. J Appl Anim Welf Sci 2002, 5, 29-42, doi:10.1207/S15327604JAWS0501_3.

Line 89: how was time from trauma estimated? Because this variable is important, any insight into how this information was obtained from the caregiver and recorded in the medical record is helpful.

Authors: we tried to clarify better how the time from trauma was estimated, and we added it in the materials and methods section (lines 135-137 _REV1, lines 126-128 _amended). We know that such definition might have led to over- or under-estimation of the actual traumatic event timing, but we decided to standardize and objectively quantify this variable as much as we could.

Line 91 & 108: both list collecting data on the clinical re-evaluation at the 3 time points.

Authors: we listed the clinical data collected at the three time points. (lines 113-116 _REV1, lines 103-106 _amended)

Line 92: Exclusion would also include cats without all of the data included in the medical record. Are these data points listed above typically included in the records? Please add to text for clarity as lack of these data points can result in cats being excluded.

Authors: the three data points are scheduled with the veterinary public service for all unowned patients undergoing orthopedic treatments at our institution. We added the exclusion criteria as requested. None of the cats included in the manuscript lacked these information. (line 119 _REV1, lines 109-110 _amended

Line 94-5: were all cats routinely testing for these viruses? Please include in text.

Authors: yes, all unowned cats are routinely tested for these viruses. We added in the text as requested. (lines 120-121 _REV1, lines 111-112 _amended)

Line 96: how was age estimated? Please include this in the text.

Authors: we included in the text. (lines 131-133 _REV1, lines 122-124 _amended)

Line 97: please include which BCS was used in the text (at least that it was the 9 point one).

Authors: added in text. (line 134 _REV1, line 125 _amended)

Line 110-11: so, these evaluations were just scored as improved or not? I don’t see any descriptive data about these 3 time periods in the results. Please add.

Authors: yes, we just scored the re-evaluations as reported. We added the descriptive data in the results section. The 90-days re-evaluation is reported directly as the CFR. (lines 293-296 _REV1, lines 282-285 _amended).

Line 140-2: is this first score not adapted from other work? Please clarify in the text. Same question for the abdominal score.

Authors: both the skeletal and abdominal scores were developed by the authors (two experienced radiologists – J.B., D.D.Z – and one experienced clinician – F.F.), and applied by only one of the authors, an experienced radiologist. We added in the text as requested. (lines 172-174, 181-182 _REV1, lines 163-165, 172-173 _amended)

Lines 161-2: this is how a variable is defined and should be included above where the variables are listed (and defined as needed).

Authors: we added the precise list of independent variables as predictors of the outcome as requested in the statistical analysis section. (lines 203-211 _REV1, lines 194-202 _amended)

Line 162: what analyses were used? Are these what is listed in lines 169-70? Please order this whole section by descriptive statistics, univariable statistics, multivariable statistical analyses and for the latter two include the list of independent variables and dependent variables.

Authors: we thank the reviewer for the precious advice. We tried to re-order and better define how statistics was performed.

How were the variables included in the model—all put in and then just reporting on which were significant still? Please add to text.

Authors: yes, this was made for the multivariable analysis, and not for the univariable. We specified in the text as requested.

How well did the logistic models fit? Please include some information about how this was evaluated here and, in the results, add what was found.

Authors: we added all the information requested, as well as the p-value of the whole multivariable nominal logistic model as an estimate of the model fit. (lines 233-234, 354-355, 357-358, 381 _REV1, lines 224-225, 322-323, 325-326, 349 _amended)

Why wasn’t a multinomial logistic regression used for the 4-category outcome variable?

Authors: it was actually done, probably not well highlighted, so we separated in a different paragraph (lines 384-85 _REV1, lines 352-352 _amended)

Line 185: were any excluded due to missing data in the medical records? Thank you for including this information for context during the time period!

Authors: none of the patients was excluded due to missing data related to our variable of interest according to our inclusion criteria in the medical records.

Line 189-194: it would be helpful to either provide a summary by category of CFR for these 4 variables or add a mean/median/percentage as appropriate to the supplemental table.

Authors: we added the requested information in the Supplementary Table 1.

Line 202: please add that 3 cats (I think) had both acute and chronic orthopedic lesions for clear context here.

Author: added in the text as requested. (lines 260-261 _REV1, lines 251-252 _amended)

Line 230: is this cat with immediate surgery one of the 2 without imaging? Please phrase more clearly (perhaps: Of these 2 patients without imaging,).

Author: added in the text as requested. (line 282 _REV1, lines 273 _amended)

I like Table 1 and would like the variables to be aligned so that it is easy to tell which were not significant for both approaches to classification.

Authors: thank you for your indication, we aligned the variables as requested. According to Reviewer#1 indications now this is Table 2.

I would then like an additional table for the univariable and multivariable analyses which included any variable significantly associated with CFR so that much of the text on pages 6-8 would be in a table instead. It might look something like below (order of columns should be consistent throughout with order…seems to be different in supplement than in text?):

Note that I just roughly calculated the numbers and % from the supplementary tables. With these data, I get a p-value of 0.2 for the chi-square. Please double check the data and analyses.

Variable

CFR (N, row %)

No-CFR (N, %)

Panleukopenia

(N, %)

Other Death (N, %)

Univariable p-value

OR & 95% CI

Logistic regression p-value

OR and 95% CI

Body weight

 <=3.3 kg

20, 63%

5, 16%

5, 16%

2, 6%

Baseline category

--

--

--

>=3.4 kg

14, 88%

1, 6%

0, 0

1, 6%

0.047

0.9-25

0.16

n/a

Outcome

0.13

--

--

--

 Adopted*

 Sheltered

Returned

*and I’m not sure I can get the data for these 12 cells from the manuscript. Only the CFR yes/no is included.

Authors: thank you for your indication, we prepared the table as requested. Now this is Table 1.

Line 235: Please include descriptive data on the length of time to CFR…that will help others understand how long this might realistically take to reach.

Authors: we have included the specific information as requested. (lines 297-302 _REV1, lines 286-291 _amended)

Line 260: is this the final CFR assessment? Are the lines 284-5 data all information from whatever the final time point was? Is there a reason why this was not examined in a multivariable analysis?

Authors: this part has now been deleted, according to previous suggestion of the Reviewer#1. Anyhow, the CFR is considered only at the final evaluation – we now specified that only for 3 cats the CFR was deemed reached at the 60-days evaluation. Regarding the lines 284-5, the CFR and no-CFR were recorded at 90-days re-evaluation (besides the 3 cats with CFR at 60-days post-evaluation, and obviously, the cats who died before the 30-days re-evaluation). Since we just retrospectively evaluated the reaching of the CFR, we categorized the time to the last evaluation as described – as well as the 2 intermediate re-evaluations, and we did not include in the multivariable analysis since we did not evaluate the time to healing.

Line 261: what post hoc analysis? The multivariable logistic model?

Authors: we thank the Reviewer for the indication, we now specified in the statistical analysis section what statistic we refer to when saying post hoc. (line 214 _REV1, line 205 _amended)

Line 273: these two re-evaluations are nested within individual cats—if they are both significant then they aren’t strongly associated with each other. However, treating them as independent variables in the multivariable analysis (I’m assuming that is what this is?) when they aren’t really, should be addressed in the discussion.

Authors: we thank the Reviewer#1 for raising the question, but we did not actually include in the multivariable model, since they might be seen as predictors but not as factors influencing the CFR. (lines 230-231 _REV1, lines 221-222 _amended)

Line 280-1: this null result could just be included as a comment in the text as it is now.

Authors: we retained the result as suggested, adding the information about the goodness of fit of the model as for previous request of the Reviewer#1.

Supplemental tables: please make sure that the titles are stand alone and explain the context of the study and all abbreviations. For Table 1: the Time to referral is not the same as Time to treatment so please be clear how that time was defined. Similarly for Table 2 for Time to therapy and Hospitalization time (is this after therapy to outcome or disposition, I’m assuming). What does “recurrence” mean in as a complication—I’m guessing of the luxation? Please clarify in table. Table 3: these are scores from 0-10, aren’t they? So why are these all words? I think the text is useful so perhaps just change the column headings and add a total trauma radiographic score column for the actual scoring. I also don’t see a summary of outcome/disposition anywhere (sheltered, adopted, euthanized/died) so perhaps add a column to Table 1 in the supplement with what happened to each cat.

Authors: we thank the Reviewer#1 for the indication, we double checked the Supplementary Tables and revised according to the Reviewer#1 suggestion. We also changed the Table 3. For the summary of the outcome disposition, we prepared a new Table 1 in the main text, as previously suggested by the Reviewer#1.

Line 291: what post hoc analyses were performed? Is this simply the results of the Wald tests for logistic regression? Please clarify.

Authors: we clarified in the text. (line 267 _REV1, line 335 _amended)

Line 316-7: that isn’t what is stated in lines 298-300. I think determining which set of CFR analyses are most important and being clear throughout the manuscript which CRF classification was used will be critical.

Authors: we thank the Reviewer for underlying the error, and we hope we now better defined the 2- and 4- categories of the CFR, as well as the significant results. (lines 395-400 _REV1, lines 364-368 _amended)

The discussion feels a little mixed up. It feels like the paragraphs starting on line 349 is quite important for this study—the authors are seeking predictors of survival from trauma. Or perhaps the more important finding is that surgery isn’t needed all the time? Please consider the fundamental purpose of this manuscript and organize the discussion from most to least important.

Authors: we thank the Reviewer for the suggestion and we tried to re-organize the Discussion as requested.

The first paragraph does give a high-level statement of the findings but uses variables that aren’t clearly defined. Please make this more stand alone and describe what are the 30- and 60-day results (improvement in function at those time points post-treatment). Lines 278-280 list a different set of significant variables.

Authors: we corrected the incorrect list of variables, as suggested. We further tried to specify the 30- and 60-days re-evaluations results.

Line 318: this should have been described in the methods or at least referenced there. And permanent disabilities like losing a leg are generally not a limitation in returning a cat to a relatively safe outdoor location. Please clarify in the text.

Authors: as for a previous comment we tried to clarify further the CFR and we referenced it in the Materials and methods section. (lines 122-124 _REV1, lines 113-115 _amended)

Lines 324-5: here is where having the data from the actual mean/median time to CFR in the paper would be helpful. The authors could indicate if their estimate was accurate and if not, what would be a better estimate.

Authors: we included the median time to CFR in the Results section. (lines 297-302 _REV1, lines 286-291 _amended)

Line 340: this could also be selection bias where heavier cats didn’t survive (maybe they were slower) to be referred to the hospital with more severe injuries (I doubt this is true, but it is another explanation for the findings which should be mentioned).

Authors: we thank the Reviewer for the comment, we added as a hypothesis in the discussion. (lines 420-421 _REV1, lines 387-388 _amended)

Paragraph starting on line 371: it is also possible that BCS was less strongly associated with CFR than weight.

Authors: we thank the Reviewer for raising the question. Actually, the BCS was associated only with the 2-categories CFR and not at the post hoc evaluation. On the other hand, the bodyweight was associated both at the univariate, multivariable and Kruskal-Wallis test in the 4-categories CFR. Anyway, evaluating the R2 at the nominal logistic analysis it resulted of 0.26 for BCS and 0.29 for bodyweight. We think that based on the calculated strength of association and this result would not add valuable information to the reader.

Line 383-4: post hoc analysis wasn’t described in the methods please add it there. And this categorization wasn’t mentioned in methods either.

Authors: we specified in the statistical analysis paragraph what we considered post hoc. (line 267 _REV1, lines 335 _amended). We specified the categorization sacro-iliac fracture luxation within the spine region. (lines 142-143 _REV1, lines 133-134 _amended)

Line 389: TTRS was associated according to line 299.

Authors: the Reviewer is right; we changed the discussion accordingly. (lines 397, 400 _REV1, lines 365, 368 _amended)

Line 404: likely because stray cats died prior to being referred in…assuming that these references apply to owned cats.

Authors: we thank the Reviewer for raising the question, and we specified that in our sample are all unowned cats who might die prior to referral in the most severe cases. (lines 495-496 _REV1, lines 459-460 _amended)

Line 440: but confounding in this study should be able to be assessed and evaluated, shouldn’t it?

Authors: we used an improper term (“confounding”) and in light of the reviewed version we deleted this sentence.

Conclusions: many readers only look at the conclusion. Please include a bit more detail about the findings and any implications from the results about whether some cats should be treated and some euthanized (doesn’t look that way but important to state) as well as how to counsel caregivers or other veterinarians about what to expect.

Authors: we thank the Reviewer for the suggestion, and we tried to improve our conclusions. (lines 577-579, 581-585_REV1, lines 536-538, 540-544 _amended)

Comments on the Quality of English Language

In a few places the language choice makes understanding the point difficult. In most places the language doesn't get in the way of understanding. However, it would be easier to make the important points of the study with some English editing.

Authors: we performed some English editing.

Reviewer 2 Report

Comments and Suggestions for Authors

The manuscript fits well within the scope of the journal and is of sound scientific value. The Authors have investigated an interesting topic and the theme has been properly described. Objectives of the study were clearly defined.

The Introduction is written concisevely but provides sufficient background. The methods have been properly described.

Results are well presented and thoroughly discussed and data interpretation is appropriate.

The manuscript is well written, presented and discussed, and understandable to a specialist readership.

Limitations of the study are listed in the manuscript. 

Author Response

The manuscript fits well within the scope of the journal and is of sound scientific value. The Authors have investigated an interesting topic and the theme has been properly described. Objectives of the study were clearly defined.

The Introduction is written concisevely but provides sufficient background. The methods have been properly described.

Results are well presented and thoroughly discussed and data interpretation is appropriate. 

The manuscript is well written, presented and discussed, and understandable to a specialist readership. 

Limitations of the study are listed in the manuscript. 

Authors: we wish to thank the Reviewer for the time spent for evaluating our manuscript, and for the appreciation showed.

Reviewer 3 Report

Comments and Suggestions for Authors

please find comments in enclosed Word file

Comments on the Quality of English Language

well written, just minor corrections of some words

Author Response

This is a well-conceived and well written retrospective study on injured stray cats treated at the emergency service of the Veterinary Teaching Hospital, University of Milan. Diagnostics, treatment, outcome, and follow-ups are appropriate as well as an extensive statistical result analysis.

Authors: we wish to thank the Reviewer for the time spent for evaluating our manuscript, and for the appreciation showed.

Recommend the paper for publication with some minor corrections – see below:

Just two comments which should be addressed by the authors:

  • There is no information on whether the recovered stray cats, when released either for adoption or re-introduced into their habitat, were neutered or not. I assume they were not, which on one hand can be understood because of their trauma, on the other, they could have been neutered after recovery before release. I think this information should be given or at least mentioned in the Discussion, as by law in some countries (e.g. the USA), no stray animal can be adopted or set free without neutering.

Authors: we thank the Reviewer for the comment, also in Italy all cats should be neutered, prior to return to their original location or adoption. We specified it in the materials and methods section. Anyway, some patients are neutered after the discharge by the veterinary public service. (lines 149-150 _REV1, lines 140-141 _amended)

This is not a multicenter study (the public health service acted only as referral); all was done and recorded at the University of Milan; thus, this reviewer wonders why for such clinical retrospective study 10 authors sign responsible and I refer to the Journal’s editor for approving such multi-authorship of this manuscript. 

Authors:  Scientific original papers that aspire to be published in peer reviewed international journals with high impact as Veterinary Sciences require the simultaneous presence of multidisciplinary scientific skills in addition to those strictly required for clinical care activities. For the aforementioned reasons, the authors included in the manuscript have been working at various levels during these three years, i.e., some more during the clinical phases, others more during the collection, extraction and preparation of data, and others more during the proper analysis of data and preparation of the manuscript. The details of these specification are included at the bottom of manuscript according to Journal guidelines. In addition, Veterinary Sciences journal did not provide any guidelines regarding the number of authors based on the type of study. Also, this Reviewer’s comment is possible only because Veterinary Sciences journal did not perform a double blinded revision. Hence, even if it is not a multicentric study, much work has been done by any of the Authors included. We wait for the Editor to evaluate the goodness of our request of 10 Authors.

Line 74…… constitutes a burden on …

Authors: corrected as suggested. (line 92 _REV1, line 84 _amended)

Line 214 what is meant by “humane euthanasia”? isn’t it always “humane” when done by veterinarians?

Authors: we erased humane as suggested. (line 271 _REV1, line 262 _amended)

Line 231 avoid using 3rd person (him) when describing the cat’s condition – please re-phrase

Line 269 delete “some”

Authors: the phrase has been changed, as requested. (line 284 _REV1, line 275 _amended)

 “Some” deleted as suggested. (line 345 _REV1, line 317 _amended)

Line 347: I am not sure whether this hypothesis finds confirmation in physical laws

(E = m x c2) since the speed in a road accident (or fall from height) is a given value, the energy released by the impact should be less when M is less…; but this is just a thought (of a non-physicist) and is not meant as suggestion for any changes 

Authors: we thank the Reviewer for the comment, but we tried to find an explanation to our findings, as from the references we included.

Line 349 better say: “… had 4,6 times less chance to reach CFR.”

Authors: corrected as suggested. (line 423 _REV1, line 390 _amended)

Line 352 suggest deleting “… which might bear more than only orthopaedic lesions.” and simply saying … in poly-traumatised cats”

Authors: corrected as suggested. (line 425 _REV1, line 392 _amended)

Line 368: “… to further deepen the patient’s assessment.” This is a bit unclear; it might be misinterpreted as an initially insufficient assessment; what you probably mean is - additional examinations, continued diagnostics and/or clinical/metabolic monitoring …?

Authors: we thank the Reviewer for improving our clearness, we rephrased as suggested. (lines 477-478 _REV1, lines 441-442 _amended)

Line 368: “On the other hand, in our experience, the absence of post-therapy complications led to an uneventful recovery and thus to a higher probability of a good return to function.” This sentence, read out of context, seems somewhat awkward: of course, this is the general experience of every clinician in every (surgical) follow-up… ... it seems undisputable true.

Authors: Indeed, we agree. But in this sample of unowned cats with the aim of reaching a CFR we wanted to stress this point. We added “even in this peculiar feline population”. (line 480 _REV1, line 444 _amended)

Line 390 “… a sort of association …” “sort of” is not a good term (in scientific writing – it is sort of vague); was there or was there not an association?; there might be marginal or non-significant association …

Authors: we thank the Reviewer for the comment, we changed in “an association” (line 458 _REV1, line 422 _amended)

Line 412 “… in kittens with lighter body weight …”

Authors: corrected as suggested. (line 504 _REV1, line 468 _amended)

Line 415 ”… in a multi-patient environment …

Authors: corrected as suggested. (line 507 _REV1, line 471 _amended)

Line 426 instead of “shows” say “inherits”

Authors: corrected as suggested. (line 459 _REV1, line 483 _amended)

Line 435 should be “validating scales”

Authors: we meant that those scales were already validated, and we changed the phrase accordingly. (line 528 _REV1, line 492 _amended)

Line 443 say “reasons” instead of “concerns”

Authors: corrected as suggested. (lines 536-537 _REV1, line 449 _amended)

Line 456 non-profit

Authors: corrected as suggested. (line 554 _REV1, line 514 _amended)

Line 469 non significant, …

Authors: corrected as suggested. (line 568 _REV1, line 527 _amended)

Reviewer 4 Report

Comments and Suggestions for Authors

I have read and reviewed this manuscript ” Potential factors influencing complete functional recovery in traumatized stray cats with orthopedic lesions – a cohort study.” with great interest and overall, from the reviewer's perspective, it is a well planned and executed study. Overall, it is a study with a simple and easy to understand wording.

However, some points need to be clarified to achieve publication quality. I have left some comments in the hope that they will help the authors.

When writing “The sum of the aforementioned radiographic scores created a ten-point scale, named total trauma radiographic score (TTRS)”. Were the scores always given by the same operator?

When writing “The estimated time from trauma to referral was 3 (0 – 60) days. How did you assess how long the subjects had been traumatised? Who made this assessment?”

Author Response

I have read and reviewed this manuscript ” Potential factors influencing complete functional recovery in traumatized stray cats with orthopedic lesions – a cohort study.” with great interest and overall, from the reviewer's perspective, it is a well planned and executed study.

Overall, it is a study with a simple and easy to understand wording.

However, some points need to be clarified to achieve publication quality.

I have left some comments in the hope that they will help the authors.

Authors: we wish to thank the Reviewer for the time spent for evaluating our manuscript, and for the appreciation showed.

When writing “The sum of the aforementioned radiographic scores created a ten-point scale, named total trauma radiographic score (TTRS)”. Were the scores always given by the same operator?

Authors: we thank the Reviewer for raising this point, we included that always one operator performed the scoring and which one was, an experienced radiologist. (line 186-187 _REV1, line 177-178 _amended)

When writing “The estimated time from trauma to referral was 3 (0 – 60) days. How did you assess how long the subjects had been traumatised? Who made this assessment?”

Authors: according to Reviewer#1 comment, we specified that estimated time from trauma to case referral, defined as the time elapsed from the first identification from the public veterinary service of a symptomatic unowned cat to the referral at the VTHUM and we included in the materials and methods section. (lines 135-137 _REV1, line 126-128 _amended)

Round 2

Reviewer 1 Report

Comments and Suggestions for Authors

The authors have substantially improved the manuscript. A few additional minor issues and one major/minor one: the nominal logistic regression results are still unclear (major) and making it clear throughout the manuscript which results are from the binomial logistic regression and which from the 4-category nominal logistic regression (minor but related to major). See below for details.

Anywhere in the manuscript where CRF is used without listing the categories, please create an abbreviation for 2 category CRF: 2-CRF and 4 category CRF: 4-CRF or something similar.

Line 23: use “associated with” rather than “correlated to” please.

Line 51: with the edit to the prior end of line, the last phrase “the so-called feline groups” can be deleted.

Line 87-8: “previous humane interactions” is this meant to be “previous interactions with humans” (so more socialized or accustomed to people)? And “more unlikely” please add afterwards something like “especially in Italy”.

Line 124 and others: generally, we use “sex” instead of “gender” for animals.

Line 145: were cats who were returned also ear-tipped or otherwise identified as sterilized? Please add for completeness.

Line 200: “internal of external” should be “internal or external”

Lines 211-14: were these comparisons the “univariable” results in Table 1 but this paragraph is about nominal logistic regression, so I’m confused. What is the difference in the univariable p-value and the “Post hoc” p-value in Table 1? And how does that relate to this section? Please edit again for clarity.

Line 246: please add “median” between estimated and time. Same for line 260 and anywhere a summary median/mean is in the text.

Lines 263-4: If this is among all orthopedic lesions, what were the other 40% treated with? Conservative therapy? Please add to text.

Lines 267-9: I only see 2 implant failures and 1 dehiscence listed in Supplemental Table 2? Could you add to complications which cats had esophagostomy tubes? And to S (surgical therapy) a -I for internal and a -E for external fixation? That would make all the data available on each cat.

Supplemental Table 3: add to the footnote that “–“ means those radiographs were not performed (I’m assuming at the discretion of cat’s veterinarian? Please add).

Line 286: so, one of the 35 who were positive at 60 days died or became negative? Please add details. I’m also assuming that the same 34 who were positive at day 30 were still positive at day 60 re-evaluation? Please add to text.

Line 287-8: Please add that these are medians. But I’m confused, above it says that 34 cats were positive at the 30-day re-evaluation and 35 at the 60 day so how were so many only positive at a median of 93 days? Please check the data and edit for clarity in text as these data are not in any other location and are critically important—time to CFR deeply influences resources needed for each cat.

Table 1: Nice work in the framework but I don’t understand given what is in the text. The univariate p-value is that the chi-square and then the Post hoc p-value the nominal one? But then how is there only one OR or relative risk ratio. For a 4-category nominal logistic there are 3, comparing each of the independent variable’s categories to the baseline category for the independent variable. Also, please round the very large OR (> 10) to the nearest whole number. Just easier to read.

For the 4 category CFR, there would need to be additional information included (the baseline category for the dependent variable and the relative risk ratio for each of the higher body weight categories to the other 3 dependent variable outcomes). I don’t know if that should go into a supplemental table or if a statement in the results indicating that the patterns were similar to what is here—except that won’t work for variables significant only in the 4 category CRF. So, two tables will be needed, one for the 2 category CRF and one for the 4 category CRF. I hope that makes sense. From paragraph starting on line 335, age, weight and comorbidities were the only places with significant differences “among the classes”—which I assume means the 4 category CRF classes (so no difference for higher TTI for example compared to the lower TTI category when comparing the No-CRF to the CRF and the Comorbidities to the CRF and the Panleukopenia to the CRF? Or?). Nominal logistic is complicated and something is missing in the manuscript.

Line 321: this paragraph doesn’t make sense with the results in Table 1. Which CFR was this used with? And how are only 3 variables significant with TTRS when a lot more are listed in Table 2? Please correct somewhere and specify how this paragraph relates to Table 1. Finding a highly significant goodness of fit means the model doesn’t fit well so something is going on there. Either explain it all or, due to sample size, omit TTRS from the multivariable analysis and edit this paragraph.

Line 376: again, this is only for the 4 category CRF, that must be included throughout the manuscript.

Line 439: “precocious” negative…I’m assuming that this means a negative score earlier in the timeline, for example, at the 30-day re-evaluation? Please edit.

Line 466-7: this isn’t in the results, is it? And how was covariance analyzed? And which CRF?

Line 474: still “parvovirus” rather than panleukopenia.

Line 519: the small sample size is likely limiting many of these additional questions. Possibly also the lack of difference among the various categories reported in results. That needs to be stated as a limitation which could have caused a lack of statistical power, leading to non-significant results, in the discussion too.

Comments on the Quality of English Language

Still some places where language makes it unclear...I've listed most of these. But is still less compelling overall due to some additional need for editing.

Author Response

REV1

The authors have substantially improved the manuscript. A few additional minor issues and one major/minor one: the nominal logistic regression results are still unclear (major) and making it clear throughout the manuscript which results are from the binomial logistic regression and which from the 4-category nominal logistic regression (minor but related to major). See below for details.

Authors: we wish to thank the Reviewer for the time spent and the efforts in evaluating and helping us improving our manuscript.

Anywhere in the manuscript where CRF is used without listing the categories, please create an abbreviation for 2 category CRF: 2-CRF and 4 category CRF: 4-CRF or something similar.

Authors: according to the Reviewer suggestion, we specified the 2-CFR and 4-CFR in the materials and methos and used them throughout the text both in the results and discussion sections.

Line 23: use “associated with” rather than “correlated to” please.

Authors: changed according to the Reviewer suggestion (line 23, amended: line 23).

Line 51: with the edit to the prior end of line, the last phrase “the so-called feline groups” can be deleted.

Authors: changed according to the Reviewer suggestion (line 51, amended: line 51)

Line 87-8: “previous humane interactions” is this meant to be “previous interactions with humans” (so more socialized or accustomed to people)? And “more unlikely” please add afterwards something like “especially in Italy”.

Authors: The Authors thanks the Reviewer for the advice. Changed according to the Reviewer suggestion. (line 87-88, amended: line 88)

Line 124 and others: generally, we use “sex” instead of “gender” for animals.

Authors: changed according to the Reviewer suggestion throughout the manuscript.

Line 145: were cats who were returned also ear-tipped or otherwise identified as sterilized? Please add for completeness.

Authors: specified according to the Reviewer suggestion. (line 141-143, amended: line 141-143)

Line 200: “internal of external” should be “internal or external”

Authors: changed according to the Reviewer suggestion. (line 203, amended: line 203)

Lines 211-14: were these comparisons the “univariable” results in Table 1 but this paragraph is about nominal logistic regression, so I’m confused. What is the difference in the univariable p-value and the “Post hoc” p-value in Table 1? And how does that relate to this section? Please edit again for clarity.

Authors: the first p-value refer to the univariate nominal logistic regression. The post hoc analysis included the tests described in the statistical analysis section - chi-square or Fisher’s F test, Student’s t test or Mann-Whitney’s U test, and the reported p-values refer to such tests. In the Table, we changed the column name in Fisher’s F that yielded the p-value, O.R. and 95%IC. In the main text we reported the ROC curve analysis and the Student’s t / Mann-Whitney’s U results. We hope that now we have explained it clearly.

Line 246: please add “median” between estimated and time. Same for line 260 and anywhere a summary median/mean is in the text.

Authors: added median wherever deemed necessary according to the Reviewer suggestion.

Lines 263-4: If this is among all orthopedic lesions, what were the other 40% treated with? Conservative therapy? Please add to text.

Authors: we thank the Reviewer for raising this point, we explained clearer in the text (line 265, amended: line 261)

Lines 267-9: I only see 2 implant failures and 1 dehiscence listed in Supplemental Table 2? Could you add to complications which cats had esophagostomy tubes? And to S (surgical therapy) a -I for internal and a -E for external fixation? That would make all the data available on each cat.

Authors: we recorded only the complications reported – we amended Table 2 using consistent terminology according to the main text (line 267-270, amended: line 263-266). We did not record any complication related to the esophagostomy tube. We adde -I (internal) -E (external), -O (other) to surgery andfor the esophagostomy tube placement.

Supplemental Table 3: add to the footnote that “–“ means those radiographs were not performed (I’m assuming at the discretion of cat’s veterinarian? Please add).

Authors: specified according to the Reviewer suggestion.

Line 286: so, one of the 35 who were positive at 60 days died or became negative? Please add details. I’m also assuming that the same 34 who were positive at day 30 were still positive at day 60 re-evaluation? Please add to text.

Authors: specified according to the Reviewer suggestion. Even though 60 days post-treatment clinical re-evaluations were classified as positive (improvement of clinical signs compared to the initial presentation), the clinical judgment at the 30- and 60-days re-evaluations did not assume the same criteria as for the CFR (i.e., ability to freely moving and jumping, no metal status alteration and no strain to urinate or defecate and absence of difficulties to eat). (line 291-293, 294-296 amended: line 287-289, 290-292)

Line 287-8: Please add that these are medians. But I’m confused, above it says that 34 cats were positive at the 30-day re-evaluation and 35 at the 60 day so how were so many only positive at a median of 93 days? Please check the data and edit for clarity in text as these data are not in any other location and are critically important—time to CFR deeply influences resources needed for each cat.

Authors: according to the Reviewer suggestion we added in the main text “median”. As answered at the previous comment, improvement of the clinical symptomatology and the absence of complications were the criteria to assign a “positive” re-evaluation. On the other hand, the criteria to classify a patient as having been reached the CFR were the ability to freely moving and jumping, no metal status alteration and no strain to urinate or defecate and absence of difficulties to eat. We decided to wait till the 90-days in most of the cases to determine the effective CFR. (line 291-293, 294-296 amended: line 287-289, 290-292)

Table 1: Nice work in the framework but I don’t understand given what is in the text. The univariate p-value is that the chi-square and then the Post hoc p-value the nominal one? But then how is there only one OR or relative risk ratio. For a 4-category nominal logistic there are 3, comparing each of the independent variable’s categories to the baseline category for the independent variable. Also, please round the very large OR (> 10) to the nearest whole number. Just easier to read.

Authors: we thank the Reviewer for raising the unclearness of the Table 1.

We tried to improving it following also the other suggestion of the Reviewer. Table 1 now contains the results of the 2-CFR:

  • The “univariate p-value” is the p-value yielded at the univariate nominal logistic regression for the tested variable.
  • The “post hoc analysis”, now corrected as “Fisher’s F test”, include the p-value, the O.R. and relative 95%CI yielded by this statistic (Fisher’s F test).
  • In the main text, results from the ROC curve analysis, Student’s t test or Mann-Whitney’s U test are reported.
  • The multivariable p-values relates again only to the 2-CFR.
  • The O.R. >10 are already rounded at the nearest whole number (i.e., bodyweight O.R. = 4.7; TTI O.R. = 4.6; Skeletal score O.R. = 0.07; 30-days O.R. = 12; 60-days O.R. = 33).

According to the next comment we prepared Table 3 for the 4-CFR. In particular in this table, we reported only number of patients and percentage, univariate nominal logistic and multivariable p-values.

For the 4 category CFR, there would need to be additional information included (the baseline category for the dependent variable and the relative risk ratio for each of the higher body weight categories to the other 3 dependent variable outcomes). I don’t know if that should go into a supplemental table or if a statement in the results indicating that the patterns were similar to what is here—except that won’t work for variables significant only in the 4 category CRF. So, two tables will be needed, one for the 2 category CRF and one for the 4 category CRF. I hope that makes sense.

Authors: we are sorry but, as explained in the first revision round, we would not be able to apply any “baseline” category, since it is possible to obtain a cut-off value from the ROC curve analysis only with a dichotomous variable, i.e., 2-CFR. Hence, the 4-CFR evaluation would not allow to display all the information that were requested by the Reviewer in the previous revision. That is why we decided to retain the 2-CFR analysis. We prepared the two tables as requested by the Reviewer.

From paragraph starting on line 335, age, weight and comorbidities were the only places with significant differences “among the classes”—which I assume means the 4 category CRF classes (so no difference for higher TTI for example compared to the lower TTI category when comparing the No-CRF to the CRF and the Comorbidities to the CRF and the Panleukopenia to the CRF? Or?). Nominal logistic is complicated and something is missing in the manuscript.

Authors: According to the Reviewer suggestion we re-evaluated the post hoc analysis and we reported each result.

Line 321: this paragraph doesn’t make sense with the results in Table 1. Which CFR was this used with? And how are only 3 variables significant with TTRS when a lot more are listed in Table 2? Please correct somewhere and specify how this paragraph relates to Table 1. Finding a highly significant goodness of fit means the model doesn’t fit well so something is going on there. Either explain it all or, due to sample size, omit TTRS from the multivariable analysis and edit this paragraph.

Authors: we thank the Reviewer for raising the point of unclearness of this paragraph. We specified that this paragraph relates to the 2-CFR. Table 2 refer to non-significant variables, whereas this paragraph relates to the 2-CFR multivariable analysis. We specified that the variables resulting significant at the multivariable analysis have to be considered as independent predictors of the outcome, as for Valveny & Gilliver 2016 (added also in the manuscript). We added as a measure of the goodness of fit the R2, which is considered the % of the variance explained in the model, i.e., how well the predictors predict the model, as for Katz 2003 (added also in the manuscript), who also states that available goodness-of-fit tests have substantial limitations. As it is now showed, the first model explains 87% of the model, whereas removing the TTRS the model falls at 32%. Finally, we added the p-value of the lack of fitness yielded by the software (JMP) which is > 0.05 (actually 0.99, 0.53, 1.00).

Line 376: again, this is only for the 4 category CRF, that must be included throughout the manuscript.

Authors: added according to the Reviewer suggestion, and specified throughout the discussion

Line 439: “precocious” negative…I’m assuming that this means a negative score earlier in the timeline, for example, at the 30-day re-evaluation? Please edit.

Authors: changed according to the Reviewer suggestion. (line 470-471 amended: line 453-454)

Line 466-7: this isn’t in the results, is it? And how was covariance analyzed? And which CRF?

Authors: we deleted from this version. It was not evaluated in the 4-CFR but on the overall sample (Spearman’s rank correlation coefficient rs= 0.80 p < 0.0001) (line 499-500 amended: line 481)

Line 474: still “parvovirus” rather than panleukopenia.

Authors: changed according to the Reviewer suggestion. (line 506 amended: line 487)

Line 519: the small sample size is likely limiting many of these additional questions. Possibly also the lack of difference among the various categories reported in results. That needs to be stated as a limitation which could have caused a lack of statistical power, leading to non-significant results, in the discussion too.

Authors: we thank the Reviewer for underline the defect. We added in the text according to the Reviewer suggestion. (line 562-566 amended: line 543-547)

Comments on the Quality of English Language

Still some places where language makes it unclear...I've listed most of these. But is still less compelling overall due to some additional need for editing.

Authors: we had language editing from an American English native speaking.

Submission Date

19 November 2023

Date of this review

13 Jan 2024 21:30:15